# Light Deficiency Inhibits Growth by Affecting Photosynthesis Efficiency as well as JA and Ethylene Signaling in Endangered Plant *Magnolia sinostellata*

**DOI:** 10.3390/plants10112261

**Published:** 2021-10-22

**Authors:** Danying Lu, Bin Liu, Mingjie Ren, Chao Wu, Jingjing Ma, Yamei Shen

**Affiliations:** 1Zhejiang Provincial Key Laboratory of Germplasm Innovation and Utilization for Garden Plants, Zhejiang Agriculture & Forestry University, Hangzhou 311300, China; 2019105052023@stu.zafu.edu.cn (D.L.); 2020105011010@stu.zafu.edu.cn (M.R.); 2020105011013@stu.zafu.edu.cn (C.W.); 2College of Landscape and Architecture, Zhejiang Agriculture and Forestry University, Hangzhou 311300, China; 3Department of Plant Genomics, Centre for Research in Agricultural Genomics (CRAG), CSIC-IRTA-UAB-UB, 08193 Bellaterra, Spain; liu.bin@cragenomica.es

**Keywords:** light deficiency, RNA-seq, gene regulation, *Magnolia sinostellata*, endangered species

## Abstract

The endangered plant *Magnolia sinostellata* largely grows in the understory of forest and suffers light deficiency stress. It is generally recognized that the interaction between plant development and growth environment is intricate; however, the underlying molecular regulatory pathways by which light deficiency induced growth inhibition remain obscure. To understand the physiological and molecular mechanisms of plant response to shading caused light deficiency, we performed photosynthesis efficiency analysis and comparative transcriptome analysis in *M. sinostellata* leaves, which were subjected to shading treatments of different durations. Most of the parameters relevant to the photosynthesis systems were altered as the result of light deficiency treatment, which was also confirmed by the transcriptome analysis. Gene Ontology and KEGG pathway enrichment analyses illustrated that most of differential expression genes (DEGs) were enriched in photosynthesis-related pathways. Light deficiency may have accelerated leaf abscission by impacting the photosynthesis efficiency and hormone signaling. Further, shading could repress the expression of stress responsive transcription factors and R-genes, which confer disease resistance. This study provides valuable insight into light deficiency-induced molecular regulatory pathways in *M. sinostellata* and offers a theoretical basis for conservation and cultivation improvements of *Magnolia* and other endangered woody plants.

## 1. Introduction

Magnoliaceae plants are widely favored by people for their pleasant ornamental characteristics and high scientific research values. The history of Magnoliaceae plants can be traced back to the Mesozoic Era [1]; however, the deciduous Magnoliaceae plants are now facing the danger of extinction in nature as the result of global climate change, forest vegetation renewal, and the succession of evergreen broad-leaved forest in subtropical areas [2]. The natural distributions of the endangered species of Magnoliaceae are mostly restricted to specific areas. 

For example, a number of deciduous Magnoliaceae species, such as *Magnolia stellata* [3], *M. wufengensis* [4], *M. Zenii* [5], *M. officinalis* [6], and *M. sinostellata* [7], tend to grow in coniferous community or forest gaps of broad-leaved communities, but are rarely found in the evergreen broad-leaved community. *M. sinostellata* is an endangered species with high ornamental value, which has been listed in the Red List of Magnoliaceae since 2016 [8]. It is a perennial deciduous flowering shrub that blossoms in early spring in the subtropical regions. 

Although *M. sinostellata* is a sun-loving plant, in its natural habitats, it grows under canopy shade or nearby the brook of the north slope in coniferous forest communities, and sparsely distributes in broad-leaved or mixed forest communities [7]. Canopy shade of evergreen plants in the upper layer of forest community will affect the growth of deciduous plants in the understory [9], as it alters the light intensity as well as the light quality towards a lower ratio of red/far-red light (R/FR) in the forest community [10,11].

Shading is recognized as an important abiotic stress that affects the growth and development of shade-intolerant plants [12]. Under shading conditions, the hypocotyls and petioles of Arabidopsis were significantly elongated, and leaf lamina growth was inhibited [13]. Leaf expansion in *Glycine max* and *Zea mays* grown under shade was inhibited, which ultimately affected their seed yield [14,15]. Vegetable seedlings of tomato, cucumber, and eggplant often grow excessively under weak light conditions, leading to decreased yield in cultivation [16]. 

Similarly, shading increased the vegetative growth but repressed the production of lemon trees [17]. In ornamental plants, shading caused leaf chlorosis in tea [18] and faded flower color of *Paeonia lactiflora* [19]. As the severity of shading increased, the leaves of *Tetrastigma hemsleyanum* and *Salvia officinalis* turned yellow due to decreased chlorophyll content [20,21]. Similarly, the leaf yellowing was also observed in deciduous *Acer pseudoplatanus*, *Fagussylvatica*, *Tilia cordata,* and coniferous *Abies alba* grown under shade in arbor [22]. However, the shade response phenotype of Magnoliaceae plants remains unclear.

Light changes under canopy shade can be sensed by phytochromes (PHY) [23], which sense changes in both light intensity and quality [24]. In Arabidopsis, the phytochrome family includes five members (phyA–phyE), among which phytochrome B plays a dominant role in shade avoidance responses including leaf expansion, stem elongation, and early flowering [25,26]. Changes in light quality with a low ratio of R/FR induce shade escape responses such as rapid stem elongation and increased apical dominance, which enable plants to reach above the upper vegetation to absorb sun light [27]. 

Low light intensity alters chloroplast ultrastructure and photosynthetic metabolism [28]. Due to inadequate energy supply, long term exposure to low light conditions can limit growth in plants and even cause cellular damage in shade sensitive plants [29,30]. The primary effect of canopy shade on understory plants is through the weakening of their photosynthesis capacity [9]. Light-harvesting antenna proteins have vital functions in the primary process in photosynthesis, which enables plants to perform photosynthesis under extreme low light environment [31]. 

Under low light intensities, to maximize photosynthetic efficiency, the size of the light harvesting antenna of the plants is often considerably larger than that grown under normal light [32]. However, the enlargement of the light-harvesting antenna is normally restricted within a certain size and, as a result, the energy generated by the photosynthesis reaction center is also limited under long term shading [32,33]. Due to the insufficient energy supply, carbohydrates transformation in the Calvin cycle was blocked under weak light [34]. 

The activity and content of ribulose 1,5-bisphosphate carboxylase/oxygenase (Rubisco) were significantly reduced under shade in *Hordeum vulgare* [35], which catalyze carbon assimilation in plants [36]. The impaired photosynthetic carbon assimilation would influence multiple metabolism pathways in plants, including starch and sucrose metabolism, amino acid metabolism, and other secondary metabolism pathways, as found in *Camellia sinensis* [37,38]. As the plant seedlings grow, the demand for light intensity increases. Consequently, the suboptimal light intensity inevitably affects the growth and reproduction of plants [39].

To cope with environmental stresses, the plants have evolved elaborate response mechanisms [40]. Phytohormones are known to have pivotal roles in regulating stress responses in plants [41]. Under low light conditions, ethylene production and signaling was induced to increase stress tolerance in Arabidopsis and rice [42,43], but its high level of release could increase the sensitivity to stress and even lead to plant death [44]. In addition, jasmonic acid (JA) is also known as plant stress hormones in higher plants [45]. JA signaling and its content accumulation are often recruited to counteract abiotic stress in tomatoes and rice [46,47,48]. 

JA interacts with ethylene signaling pathways to regulate abiotic stress tolerance and may act as a core signal in such an intertwined network [49]. Various studies found that light deficiency can alter stress tolerance and disease resistance in plants [50,51,52,53,54,55,56]. However, its molecular mechanisms in plants are complex and unclear. Numerous transcription factors (TFs) are involved in regulating molecular response in plants [57]. TIFY family genes are plant-specific and responsible to multiple abiotic stresses, most of which are stress inducible and able to enhance plant stress tolerance [58], as exemplified in *Citrullus lanatus* [59], *Triticum aestivum* [60], and *G. max* [61]. 

The mitochondrial transcription termination factor (mTERF) family was also found as essential for plant adaptation or tolerance to abiotic stresses by regulating transcription, translation and DNA replication in mitochondria in plants [62,63]. Furthermore, plant resistance (R) genes encode immune receptors that recognize specific avirulence (Avr) genes of pathogens [64], the expression levels of which are tightly associated with disease resistance and could be induced by environmental stress [65]. To our knowledge, the responses of TIFYs, mTERFs, and R-genes to weak light have not been investigated. Further, the regulatory pathways of molecular responses to light deficiency in *M. sinostellata* remain obscure. 

Previous studies have confirmed that weak light is the main factor that affect the distribution and growth of deciduous magnolia in the understory of forest community [5,7]. The light environment under canopy shade can be quite complex, containing low light intensity and inconsistent light quality [66,67]. To reduce the confounding factors, many studies focused on the impacts of low light intensity on a number of plant species, such as soybean [68], rice [69], *Halimium halimifolium* [70]. Accordingly, in this study, we mainly focused on the effect of shading caused light deficiency on *M. sinostellata* and black shade net was used to create light deficiency conditions. 

In the preliminary research, we found that low light intensity promoted leaf abscission in *M. sinostellata*, which is distinct from the response of model plants, such as Arabidopsis and rice. Photosynthesis efficiency analysis revealed that light deficiency altered photosynthesis systems related parameters. Comparative transcriptome analysis in *M. sinostellata* leaves with and without weak light treatment led to the identification of numerous core low-light responsive pathways, TFs, and R-genes. This study may offer a new insight into the light deficiency response mechanism, which forms a theoretical basis for the protection and conservation of *M. sinostellata* and other endangered woody plants.

## 2. Results

### 2.1. Light Deficiency Inhibited Growth of M. sinostellata

Significant morphological changes in *M. sinostellata* seedlings under light deficiency treatment (LT), relative to the control (CK) seedlings, were observed (Figure 1A). The CK leaves of *M. sinostellata* were fully expanded, and no significant changes were observed in these seedlings at day 1. After being subjected to low light conditions for 5 d, seedlings began to wilt, and around 5% of leaves were shed (Appendix A). By 15 d of light deficiency treatment, severe leaf abscission was observed, 17% of leaves fell. After 20 d, most of the leaves were found to face downward, and approximately 28% of leaves were shed. 

Furthermore, the whole plants began to show evident phenotypic damage, which worsened as the light deficiency treatment progressed. After 25 d of the treatment, 34% of leaves were shed, and some leaves were completely dried up. After 30 d of the light deficiency stress, 40% of the leaves were shed. Such morphological changes confirm that *M. sinostellata* is highly sensitive to light deficiency. To investigate the potential changes in carbon assimilation as caused by light deficiency treatment, a number of photosynthetic parameters were measured in the *M. sinostellata* leaves, including net photosynthetic net (P_n_), intercellular CO_2_ concentration (C_i_), stomatal conductance (G_s_), and transpiration rate (T_r_). 

Under weak light, P_n_ initially increased by 20% after 10 d shade treatment, which were reduced to 60% of that of the controls after 30 d light deficiency treatment (Figure 1B). C_i_ (Figure 1C) also showed an initial increase prior to dropping at 25 d, while G_s_ (Figure 1D) and T_r_ (Figure 1E) consistently decreased. The light utilization efficiency (LUE) (Figure 1F) and water utilization efficiency (WUE) (Figure 1G) were also analyzed, which showed similar trends in response to weak light, both of which peaked at 10 d and then deceased. 

Rubisco activity was peaked at d5 before its sharp reduction (Figure 1H). Chlorophyll fluorescence parameters were also affected by light deficiency stress throughout the experiment (Figure 1I–P). The maximal fluorescence (Fm), maximal quantum yield of PSII (Fv/Fm), excitation energy capture efficiency of PSII (Fv’/Fm’), photochemical quenching (qP), active PSII reaction centers (Fv/Fo) and non-photochemical quenching (NPQ) values all decreased throughout the experiment. Fv’/Fm’ values rapidly decreased until 10 d, and then dropped down slowly thereafter. qP declined throughout the experiment, while the initial fluorescence (Fo) values increased consistently. The yield of PSII photochemistry (ΦPSII) value decreased until 5 d, followed by a sharp increase from 5 d to 15 d, before significantly dropping thereafter. 

### 2.2. Summary of Transcriptome Assembly and Function Annotation in M. sinostellata

Based on the results of the photosynthesis analysis, the samples of d 0 (mixed sample of CK and LT), d 5, and d 15 in both CK and LT were selected for transcriptome sequencing. A total of 15 samples in five groups (CK-D0, CK-D5, CK-D15, LT-D5, and LT-D15) were mixed equally, and used for the full-length transcriptome sequencing, which obtained a total of 50.13 GB data and 14,653,022 subreads. The length of subreads varied from 3420.95 bp to 188,350 bp (Appendix A). After de-redundancy, 246,481 unigenes were obtained in *M. sinostellata* with a total length of 270,112,156 bp, and the GC content was 43.97% (Appendix A). 

BUSCO was used to evaluate the completeness of transcriptome assembly, which showed that full-length transcriptome of M. sinostellata was comprised of 88.78%, 4.95%, and 6.27% of the complete, fragmented and missing BUSCOs, respectively (Appendix A). All the unigenes were blasted against the seven public databases for functional annotation (Appendix A). 173,103, 146,820, 128,216, 135,136, 128,718, 107,462, and 138,676 unigenes were identified in the database of Nr, Nt, Swissprot, KEGG, KOG, Pfam, and GO, respectively, which became the basis for the functional annotation of a total number of 191,343 unigenes. 

The high-quality full-length consensus sequences obtained by full-length transcriptome sequencing were employed as the reference gene set for M. sinostellata. To further elucidate the shade responsive patterns of M. sinostellata, de novo transcriptome sequencing was performed on the 15 samples separately and a total of 697.63 M original reads were obtained (Appendix A). When the clean reads obtained by the second-generation transcriptome sequencing were aligned to the reference gene set by Bowtie2, a total of 181,902 genes were detected in this de novo transcriptome sequencing. 

The mapped ratios were varied from 73.76% to 86.99% with the mean of 80.49% (Appendix A). A box plot of the gene expression in FPKM value as calculated using RSEM illustrates the overall distribution of gene expression in each sample (Appendix A). A sample PCA map was generated by analyzing all the 15 samples by dimensionality reduction method (Appendix A), which shows a high level of correlation among the three biological replicates in five groups.

### 2.3. Identification of Differentially Expressed Gene in M. sinostellata

In total, 11,850, 12,320, 7165, and 15,389 DEGs were detected in CK-D0-vs-LT-D5, CK-D5-vs-LT-D5, CK-D0-vs-LT-D15, and CK-D15-vs-LT-D15 comparison group, respectively (Appendix A). Following the removal of overlapping DEGs detected in the four comparison groups, a total of 22,433 DEGs for light deficiency response were identified based on strict criteria (Fold change > 4 and *p* < 0.05). A Venn diagram showed that 3309 DEGs were significantly expressed throughout the treatment (Appendix A). Among the 22,433 DEGs, GO analysis indicated that the top five enriched GO terms were directly related to photosynthesis components (Figure 2A), which are all photosynthesis and thylakoid related terms (GO:0009765, GO:0009579, GO:0009522, GO:0034357, and GO:0009521). 

KEGG analysis showed consistent results with GO analysis. The top five enriched KEGG pathways were all associated with photosynthesis, carbohydrate metabolism or other secondary metabolism, among which ‘Photosynthesis—antenna proteins’ was the most enriched KEGG pathway, followed by ‘Phenylpropanoid biosynthesis’, ‘Glycine, serine and threonine metabolism’, ‘Cyanoamino acid metabolism’, and ‘Starch and sucrose metabolism’ (Figure 2B). Taken together, these results strongly suggest that photosynthesis, carbon fixation and other secondary metabolisms play important roles in response to light deficiency in *M. sinostellata.*

### 2.4. Expression Patterns of DEGs Involved in Photosynthesis Altered under Light Deficiency

To further investigate the effects of light deficiency on the photosynthesis capacity of *M. sinostellata*, the expression patterns of DEGs involved in photosynthesis-related pathways were analyzed. Under the low light condition, most of the genes that were predicted to function in antenna protein pathway (ko00196) [71] were significantly down regulated (Figure 2C). In this pathway, gene expression trend analysis indicated that 336 genes were consistently down regulated, whereas three genes were up regulated (Figure 2D). Heatmap analysis showed that expression level of most genes in light harvesting complex I (LHC I and LHCA1-5) and light harvesting complex II (LHC II and LHCB1-7) were decreased under light deficiency treatment (Figure 3A), suggesting that the activity of light-harvesting chlorophyll protein complex (LHC) of *M. sinostellata* was inhibited. 

Similarly, DEGs that are related to the photosynthesis pathway (Ko00195) were also mainly down regulated during light deficiency conditions, among which 568 DEGs were significantly down regulated, whereas 26 genes showed a trend of significant upregulation (Figure 2E,F). Compared with CK, the genes involved in the PSI, PSII, including Cytochrome b6/f complex, photosynthetic electron transport and F-type ATPase, were mostly down regulated under light deficiency treatment, consistent with the reductions in photosynthesis efficiency (Figure 3B). 

In the carbon fixation pathway (Ko00710), the expression of 328 genes declined, whereas 31 genes were dramatically up regulated (Figure 2G,H). The significant decrease in the transcription level of the key genes involved in Calvin cycle, including RbcLs, RbcSs, PGAs, GADPHs, GAPAs, and TPIs, may indicate that the carbon assimilation efficiency and further sucrose and starch metabolism were depressed in the shade treated *M. sinostellata* plants (Figure 3C). 

### 2.5. Phytohormonal Changes in Leave of M. sinostellata under Low Light

Transcriptome data analysis showed that light deficiency mainly affected ethylene and jasmonic acid signaling pathways. Under light deficiency treatment, the ethylene signaling promoting factors, such as EIN3 and ERFs, were significantly up-regulated by weak light, indicating the activation of ET mediated signaling pathway (Figure 4A). A number of key genes related to JA signaling were down regulated by weak light, among which JA signaling inhibitor COI1 was the most significantly induced. 

Further, the expression of stress response regulators JAR1 and JAZ were inhibited in the JA signaling pathway, which suggests that the JA-mediated stress response pathway was suppressed by low light (Figure 4B). To further verify whether weak light affects phytohormone accumulation, the concentrations of endogenous ACC (ethylene precursors) and JA were measured during the experiment. After 15 d light deficiency treatment, the endogenous ACC level increased from 83.3 to 153.5 ng/g, Figure 4C. In contrast, the endogenous JA level dropped from 19.8 to 13.1 ng/g at 15 d after experiment (Figure 4D). The alteration in the expression patterns of the above mentioned core genes involved in plant hormone signaling were verified using qRT-PCR, which were consistent with the transcriptome analysis (Figure 4E).

### 2.6. Light Deficiency Affected Stress-Related Transcription Factors in M. sinostellata

Given that low light intensity can impact stress tolerance in various plants, such as Calamus viminalis, Anoectochilus roxburghii, and Leymus chinensis [50,51,52], and light deficiency also weakened the resistance of *M. sinostellata* [53], stress response TFs were identified and analyzed in a genome wide range. TIFY and mitochondrial transcription termination factors (mTERFs) are related to stress response and have important roles in stress tolerance in plants [72,73]. Seven MsTIFYs were identified from the *M. sinostellata* transcriptome, and their physicochemical characters are listed in Appendix A. 

Based on FPKM values of MsTIFYs, heatmap analysis showed that MsTIFYs were all consistently down regulated during light deficiency treatment, except for MsTIFY6, which showed slight upregulation after 15 d (Figure 5A). To understand the evolutionary relationship of MsTIFYs with its orthologs in other species, a phylogenetic tree consisting of 39 TIFYs were constructed. As shown in Figure 5B, the 39 TIFY proteins could be classified into seven subgroups, including TIFY, ZML, PPD, and JAZ I–Ⅳ, among which MsTIFYs were localized in the ZML and JAZ I–Ⅳ subgroups, among which MsTIFY3 and MsTIFY9 were clustered in subgroup JAZ I; MsTIFY5a and MsTIFY5b were localized in the JAZ III subgroup; and MsTIFY10a, MsTIFY10b, and MsTIFY6 were clustered in the subgroup JAZ II, ZML, and JAZ Ⅳ, respectively. The subgroup TIFY and PPD comprised AtTIFYs and PtTIFYs only. 

Seven MsmTERFs were detected among the 22,433 weak-light responsive DEGs (Appendix A). The prediction of protein location showed that MsmTERF1, MsmTERF3, and MsmTERF12 were localized in the nucleus. MsmTERF4, MsmTERF6, and MsmTERF10 were predicted to be located in chloroplasts. Interestingly, the MsmTERF7 protein was predicted to function in the cell membrane or chloroplasts. The expression levels of all the seven MsmTERFs declined during low-light treatment, among which the decrease of MsTERF1, MsTERF3, MsTERF10, and MsTERF12 were significant (Figure 5C). 

Based on these results, we could assume that MsmTERFs may have participated in low-light response regulation in *M. sinostellata*. The phylogenetic analysis of the 73 mTERF proteins indicated that they could be divided into five subgroups (subgroups I–Ⅴ) (Figure 5D). Among seven MsTERFs, six MsTRRFs were clustered into subgroup II, and only MsmTERF7 was clustered in subgroup Ⅳ. Subgroup I and III only contain AtmTERFs and ZmmTERFs. Interestingly, subgroup Ⅴ was clustered by 18 ZmmTERF protein sequences. The seven MsmTERFs were all clustered with its *A. thaliana* ortholog. Accordingly, these mTERF genes might have similar functions to their corresponding *Arabidopsis* orthologs.

### 2.7. Low Light Treatment Altered Expression Pattern of R-Genes in M. sinostellata

Accumulating evidence showing that changes in light condition can alter disease resistance in various plants [54], such as coffee, tomatoes, Acer rubrum, and Prunus serotina [55,56,74]. To preliminary explore the effect of light deficiency on disease resistance related molecular mechanism in *M. sinostellata*, R-genes were identified, and its expression pattern under light deficiency was analyzed. The 22,433 DEGs were blasted with the plant resistance gene database (PRGDB, http://prgdb.crg.eu/, access date 30 December 2020), and 1086 R genes belonging to 13 families were functionally annotated (Table 1). Among them, 644 were down-regulated, and 442 were up-regulated, showing that the disease resistance ability in *M. sinostellata* may have been somewhat compromised under shading (Figure 4E). 

Among five Mlo-like class of R genes, four were significantly down-regulated, whereas one was up-regulated in response to low light treatment. It is interesting to note that the expression level of most R genes belonging to the CN, N, NL, RLP, and RLK-GNK2 were drastically reduced. All of the R genes in RLK and RPW8-NL classes were down regulated. In contrast, most R genes belonging to the T, TN, and other classes were up regulated, indicating that their activation was triggered by light deficiency. In both CNL and TNL classes, almost half of the R genes were up-regulated, whereas the other half were down-regulated.

## 3. Discussion

Canopy shade is a major abiotic stress that affect the growth and development of endangered plants in the understories of forest community [37,75]. In this study, we found that light deficiency had exacerbated leaf abscission, and negatively impacted photosynthesis and phytohormone signaling of *M. sinostellata* through a comprehensive analysis of transcriptome data. 

Phenotype reflects growth and development condition of plants [76]. During the experiment, mature leaves of *M. sinostellata* gradually wilt and leaf abscission aggravated after being subjected to light deficiency treatment, which is distinct from many other plants, confirming that *M. sinostellata* is highly sensitive to light deficiency. When grown under weak light, elongated hypocotyls and shoots of *Lycopersicon esculentum* [77], *Arabidopsis* [78,79] and *C. sativus* [80] were observed. Excessive growth and elongated phenotypes were also observed in tomato, cucumber, eggplant, and lemon [16,17]. 

In addition, leaf chlorosis and senescence were reported in rice [81], *Camellia sinensis* [82] and *Acer pseudoplatanus* [22] under weak light, which observed in *M. sinostellata*. However, to our knowledge, leaf abscission exacerbated by light deficiency has not been observed in any other plants, which severely impaired the growth and development of *M. sinostellata* seedlings.

Photosynthesis is essential for maintaining plant growth and development [83], and it was significantly impacted by light deficiency in *M. sinostellata*. Under low light conditions, light captured by antenna proteins were limited. Light-harvesting antenna proteins have vital functions in photosynthesis, which enables plants to perform photosynthesis under extreme low light environments [31]. When under a short period of low light stress, the size of the light harvesting antenna increased to absorb more light [32] and the expression level of *LHC* genes in *C. sinensis* and soybean were significantly induced [28,82]. Nevertheless, the enlargement of the light-harvesting antenna is limited [33] and the expression of *LHC* genes in *Arachis hypogaea* showed a downward trend under long-term light deficiency [11]. 

Similarly, the expression of *LHCA1-5* and *LHCB1-6* were also decreased in *M. sinostellata* under long term light deficiency (Figure 2C,D, and Figure 3A). Photosynthetic parameters are commonly used as indicators of plant photosynthesis ability [84]. During the experiment, *P*_n_ was observed to decline after reaching its highest value. *G*_s_ and *T*_r_ consistently declined, whereas *C*_i_ significantly increased in *M. sinostellata* (Figure 1B–E). Similar results were reported in Sweet Pepper [85], *Zea mays* [86], and *Castanopsis kawakamii* [87], showing that the photosynthesis was inhibited under low light stress. Fv/Fm is a vital indicator of photosynthetic capacity [88] and it declined under light deficiency in *M. sinostellata* (Figure 1L), indicating that photosynthesis was impaired. 

The reduction in the values of other parameters such as Fv/Fo, NPQ, qP, and ΦPSII (Figure 1I–P) were also consistent with previous findings in *Cucumis sativus* [89] and wheat [90]. These results suggest that electron transport chain in PSI and PSII could be blocked during light deficiency. Genes encoding the components of both PSI and PSII were repressed in *M. sinostellata* during experiment (Figure 2E,F, and Figure 3B). In most plants, photosystems are sensitive to the change in light environment and their activity could be suppressed under low light stress [91]. 

Rubisco activity declined after an initial rise during the light deficiency treatment (Figure 1H). Together with the reductions in the expression of *RbcL* and *RbcS* in response to light deficiency, the reduction in Rubisco activity is in line with previous findings in *C. sativus* [92]. A number of other genes involved in the Calvin cycle, including *GADPA*, *GAPA* and *TPI*, were also suppressed by shade (Figure 2G,H, and Figure 3C). Taken together, it is reasonable to say that carbon assimilation in plants could be depressed due to the down-regulation of a suite of genes involved in carbon fixation, which directly inhibits plant growth.

Phytohormones are essential for coordinating various signaling pathways in response to abiotic stress in plants [43]. In addition to the stress response and plant growth, phytohormones can also regulate leaf abscission [93,94], which were observed in *M. sinostellata* in this study (Figure 1A). The increase in ethylene content can promote leaf abscission [95,96]. 1-Aminocyclopropane-1-carboxylic acid (ACC) is the direct precursors of ethylene, and its content is positively correlated with ethylene [97]. The ACC level *in M. sinostellata* increased in response to light deficiency (Figure 4C), indicating that the content of ethylene also increased under weak light. 

The core regulators of ethylene signaling ethylene-insensitive 3 (EIN3) and ethylene response factors (ERF) were mostly induced under shading in *M. sinostellata* (Figure 4A). ERF functions downstream of EIN3 and drives ethylene-induced senescence [98]. Further, ethylene can facilitate leaf abscission by weakening the cell walls in the abscission zone [94]. The activation of JA accumulation and signaling triggers the plant stress response and enhances stress tolerance [99]. The increase in JA content under abiotic stress can enhance plant resistance [100], while its reduced content under long term stress increases stress sensitivity [101]. 

In this study, the level of endogenous JA decreased in *M. sinostellata* leaves under light deficiency (Figure 4D). In addition, the expression of *JAR1* decreased under low light, which interacts with coronatine-insensitive protein 1 (COI1) and then leads to the degradation of JAZ proteins. The down regulation of *JAZ* is an indication of the weakening of stress resistance [72]. As *MYC2* is key transcription activator of JA-Ile/COI1 signaling [102], its downregulation under light deficiency is perhaps not surprising (Figure 4B). Collectively, the exacerbated leaf abscission observed in this study could be explained by the concerted regulation of ethylene and JA signaling pathways.

Various studies found that low light intensity can impact disease resistance in plants, and many works proved that plants reduced stress tolerance under light deficiency [50,51,52,53,54,55,56]. Moreover, our previous study also found that light deficiency impacted stress tolerance in *M. sinostellata* [64]. To explore the mechanism at the molecular level, stress-related TFs and R genes were identified and analyzed. Stress responsive transcription factors TIFY and mTERF are closely associated with defense and stress response [62,103]. Most TIFY family genes are stress inducible and able to enhance plant stress tolerance by its high expression [58,61,104]. 

In this study, the expression of all seven *MsTIFY*s were regulated by light deficiency (Figure 5A), suggesting that its function was suppressed under long term light deficiency. Another stress-responsive TF family mTERF was also reported to regulate plant development and various stress responses [63,105]. Down-regulation of *mTERF*s would impair chloroplast or mitochondria development [62]. Mutants of *AtmTERF9* showed altered response to multiple abiotic stresses [106]. The defective mutants of *mTERF6* and *mTERF10* in Arabidopsis were hypersensitive to multiple abiotic stresses, while their overexpression could increase stress tolerance [63,105]. The consistent decline in the expression levels of the seven MsmTERFs identified in *M. sinostellata* (Figure 5C,D) is in agreement with previous findings in *Z. mays* [73]. R-genes play pivotal roles in restricting pathogen invasion and triggering plant defense responses [107]. 

The R-genes are classified into five main groups according to their conserved domains and motifs [108]. Due to the high costs of maintaining R-protein-dependent expression, expression levels of R genes are tightly regulated [109]. The expression pattern could be altered by both biotic and abiotic stresses [110,111]. The increased expression of R genes could enhance immunity to bacterial pathogens in plants [112]. The alterations in the expression patterns of a large number of *M. sinostellata* R-genes discovered in this study suggest that the overall resistance capacity to various pathogens might be affected (Figure 5E), which warrants further studies.

The preliminary study suggested that *M. sinostellata* was hypersensitive to low light intensity and weak light could severely impact on photosynthesis, phytohormone signaling, expression of stress related TFs, and R-genes of *M. sinostellata*.

## 4. Materials and Methods

### 4.1. Plant Materials and Shade Treatments

The *M. sinostellata* seedlings were collected from the Lin’an district, Hangzhou in Zhejiang province, China. Throughout the experiment, these seedlings were placed in an artificial climate room (photosynthesis active radiation (PAR) of 648 μmol·m^–2^·s^–1^, 14 h photoperiod, temperature 25 °C, humidity 40–60%) in Zhejiang Agriculture and Forestry University. In order to simulate shade-caused low light intensity conditions, seedlings in the treated group (light deficiency treatment, LT) were placed in the shade set-up, which was built using black shade net (25% of full light, PAR of 162 μmol·m^–2^·s^–1^, R/FR ratio: 1.09) and several bamboo poles (Appendix A). Seedlings in the control group (control, CK) were not shaded (100% of full light, PAR of 648 μmol·m^–2^·s^–1^, R/FR ratio: 1.10). 

The illumination intensities in the control group and treated group were measured in luminous flux (LUX) with a digital luxmeter (ZDS-10, Shanghai Jiading Xuelian Instrument Co., Ltd., Shanghai, China). Light intensity was converted from LUX to PAR following methods by Chen [113]. R/FR ratios under different conditions were measured by using a NIR spectrometer (Avaspec-HS-TEC, Avantes, The Netherlands). The data of light intensity and quality in experimental or natural conditions is provided in Appendix A. All other experimental conditions were maintained the same for both LT and CK. Each group comprised three replicates. Leaf samples were collected from the seedlings in LT and CK groups at 0, 1, 5, 10, 15, 25, and 30 days (d) and stored at −80 °C for further experiments after being snap frozen in liquid nitrogen until further experiment. Each sample was collected from three seedlings, and each collection was repeated three times as biological replicates.

### 4.2. Measurement of Photosynthetic Parameters

The photosynthetic parameters, including the net photosynthetic rate *(P_n_*), intercellular carbon dioxide concentration (*C_i_*), stomatal conductance (*G_s_*), and transpiration rate (*T_r_*) were measured between 9:00 and 11.30 a.m. using a LI-6400 photosynthesis analyzer (LI-COR Biosciences, Lincoln, NE, USA). The water-use efficiency (WUE) and light-use efficiency (LUE) were calculated according to the formulas: *WUE* = *P_n_*/*T_r_*; *LUE* = *P_n_*/PAR (photosynthetically active radiation). The parameters of the photosynthesis analyzer were set as follows: CO_2_ concentration at 380 µmol·mol^–1^; airflow rate at 500 µmol·s^–1^; block leaf temperature at 25 °C; photosynthetic photon flow density (PPFD) at 800 µmol·m^−2^·s^−1^. All the measurements were performed in triplicate.

### 4.3. Measurement of Chlorophyll Fluorescence Parameters

The chlorophyll fluorescence parameters were determined using a leaf chamber with a red/blue light source in the LI-6400 portable gas exchange system (Li-Cor, Lincoln, NE, USA). Before the initial fluorescence intensity (Fo) determination, leaves were dark-adapted for 30 min. Subsequently, the maximum fluorescence (Fm) was induced and measured by applying a flash of saturating light (6000 μmol·m^−2^·s^−1^, 0.7 s). When measuring Fo and Fm, the PPFD was set at 0 µmol·m^−2^·s^−1^. After fully light-adaption, the steady-state fluorescence level (Fs) and maximal fluorescence (Fm’) were measured with the PPFD set at 800 µmol·m^−2^·s^−1^. The maximal photochemical efficiency (Fv/Fm), actual photochemical efficiency (ΦPSⅡ), non-photochemical dissipation of absorbed light energy (NPQ), and the coefficient for photochemical quenching (qP) were calculated according to the following formulas [86,89]: Fv/Fm = (Fm − Fo)/Fm; ΦPSII= (Fm’ − Fs)/Fm’; qP = (Fm’ − Fs)/(Fm’ − Fo’); and NPQ = (Fm − Fm’)/Fm’. Each measurement comprised three replicates.

### 4.4. Determination of Rubisco Activity

Fresh leaf samples of *M. sinostellata* were pooled to measure and calculate the Ribulose diphosphate carboxylase/oxygenase (Rubisco) activity. The Rubisco activity was determined according to the instruction of the Rubisco assay kit (Beijing Solarbio Science & Technology Co., Ltd., Beijing, China). The Rubisco enzyme activity was measured in the unit of U/g, which represents the oxidation of 1 μmol of NADH per min. Each enzyme activity determination was repeated at least three times.

### 4.5. Transcriptome Sequencing and Analysis

Based on physiological measurement results, samples of d0 (mixed samples of CK-D0 and LT-D0), d5 (CK-D5 and LT-D5), and d15 (CK-D15 and LT-D15) with three biological replicates were collected for subsequent sequencing and analysis. To obtain full-length cDNA sequences, the total RNAs of all the 15 samples were mixed in equal quantities to construct a PacBio Iso-Seq library. The first strand cDNA was synthesized using UMI base PCR cDNA Synthesis Kit (Beijing Genomics institution, Beijing, China). 

High-quality full-length consensus sequences were obtained after using SMRT analysis suite to perform insert recognition including Reads of insert (ROI), Reads classification (Classify), and Reads clustering and correction (Cluster, Quvier), which serve as the reference gene sequences of *M. sinostellata*. Clustering de-redundancy was performed, and the transcripts were then annotated using the public databases including NR, NT, swiss-prot, COG, KEGG, and GO. The total RNAs of the 15 samples were purified, fragmented, reversed, and then synthesized into cDNAs to form double-stranded DNAs. The ends of synthetic double-stranded DNA were filled and repaired. Using specific primers to amplify the ligation product by PCR, the product was denatured into single-strands to obtain a single-stranded circular DNA library using a bridge primer. The single-stranded circular DNA library was sequenced on the DNBSEQ platform.

### 4.6. Analysis of Shade Responsive DEGs

The expression level of each transcript was normalized using the FPKM (fragments per kilobase of exon per million mapped fragments) method. The R package (edgeR v3.16) was employed to analyze the normalized gene expression level data in order to identify the differentially expressed genes (DEGs) under shading treatments. Genes that fulfilled the criteria of Log2fold change > 2 and q value < 0.05 were identified as DEGs, with a false discovery rate (FDR) < 0.05. The function of the DEGs was annotated using the Gene Ontology (GO) and Kyoto Encyclopedia of Genes and Genomes (KEGG). The phyper function in R software was employed for enrichment analysis.

### 4.7. Bioinformatic Analysis

DEGs that are involved in the light-regulated and plant hormone pathways were filtered based on GO and KEGG annotation. To identify stress-response TFs, the Getorf function in EMBOSS software was used to predict the open reading frames (ORFs) of the unigenes, from which the TF domains were searched using the Hmmsearch function. CLUSTALW was used to align the amino acid sequences of TFs, and the neighbor joining trees were constructed by using MEGA 5.0 software. The fasta format of DNA sequences of the unigenes was subjected to blast search against the plant disease resistance gene database (PRGDB) using the DIAMOND software. The R genes were filtered and obtained according to the query coverage and the identity of the blast results. Volcano plots were performed using R software. Heatmaps were generated using Morpheus (Morpheus, https://software.broadinstitute.org/morpheus, access date 12 October 2021).

### 4.8. qRT-PCR Analysis

The total RNA of the 15 leaf samples was extracted using an RNAprep Pure Plant Plus kit (TIANGEN Biotech, Beijing, China) following the manufacturer’s instructions. Electrophoretic apparatus DYY-6C (LIUYI Biotech, Beijing, China), and agarose for electrophoresis use (Sangon Biotech, Shanghai, China) was used to analyze the integrity of RNAs. The purity and concentration of the total RNAs were analyzed by a NanoDrop system (Thermo Fisher Scientific, Waltham, MA, USA). The total RNAs were converted to cDNAs using the PrimeScript™ RT Master Mix (Takara Bio Inc., Shiga, Japan). 

Premier 5.0 was employed to design oligo primers for quantitative real time PCR (qRT-PCR), as listed in Appendix A. qRT-PCR analysis was performed on LightCycler^®^ 480 II (Roche Applied Science, Penzberg, Germany) using BCG qPCR Master Mix (Beijing Baikaiji Biotechnology Co., Ltd., Beijing, China) using a program that was set with an initial denaturing at 95 °C for 30 s, which was followed by 40 cycles of 95 °C for 5 s and 58 °C for 30 s. Melting curves were generated after the end of the program from 65 °C to 95 °C with 0.2 °C increments. *M. sinostellata EF1-α* was employed as the reference gene (Forward: 5′-GATGATTCCAACCAAGCCCA -3′, Reverse: 5′-CACCCACTGCAACAGTCTGG -3′) and gene expression was determined using 2^−∆∆Ct^ method [114]. 

All the qRT-PCR analysis experiments were performed in triplicate. The bar charts of the relative expression level were generated using the Graph pad software (Graph Pad Software, San Diego, CA, USA). SPSS software version 24.0 (SPSS, Inc., Chicago, IL, USA) was employed to analyze statistical significance.

### 4.9. Phytohormone Quantification

In order to analysis the trend for change in phytohormones, leaf samples of d0 (mixed samples of CK-D0 and LT-D0) and d15 (CK-D15 and LT-D15) with three biological replicates were collected for phytohormone quantification. Approximately 500 mg of each sample was rapidly frozen in liquid nitrogen. The extraction and quantification of endogenous ACC (ethylene precursors) and JA were performed using an LC-ESI-MS/MS system (UPLC, Shim-pack UFLC SHIMADZU CBM30A system, http://www.shimadzu.com.cn/, access date 12 October 2021, Kyoto, Japan; MS/MS, Applied Biosystems, Foster City, CA, 6500 Quadrupole Trap, http://www.appliedbiosystems.com.cn/, access date 12 October 2021) by Wuhan Metware Biotechnology Co., Ltd. (Wuhan, China) [115,116,117,118,119].

## 5. Conclusions

We provided novel insights into the light deficiency response mechanism in an endangered ornamental tree species *M. sinostellata* through the analyses of transcriptome deep sequencing and photosynthesis efficiency. Under low light conditions, the intensity of light that captured by light harvesting complex was reduced. Then, the electron transport chains in PS I and PS II were affected. Due to the decline in ATP and NADPH production in the photosystem, the carbon fixation in the Calvin cycle and subsequent carbon metabolism were blocked. 

The signaling transduction and accumulation of ET were enhanced, while the content of JA was reduced in *M. sinostellata*. The expression of stress-responsive transcription TIFY and mTERF was repressed. In addition, most R-genes were downregulated under low-light conditions. Overall, light deficiency could impact on the growth of *M. sinostellata* plants through its direct effects on photosynthesis, phytohormone transduction, stress related TFs, and R-genes (Figure 6). The DEGs discovered in this study may facilitate further study on light deficiency responsive molecular mechanisms in plants.

## Figures and Tables

**Figure 1 plants-10-02261-f001:**
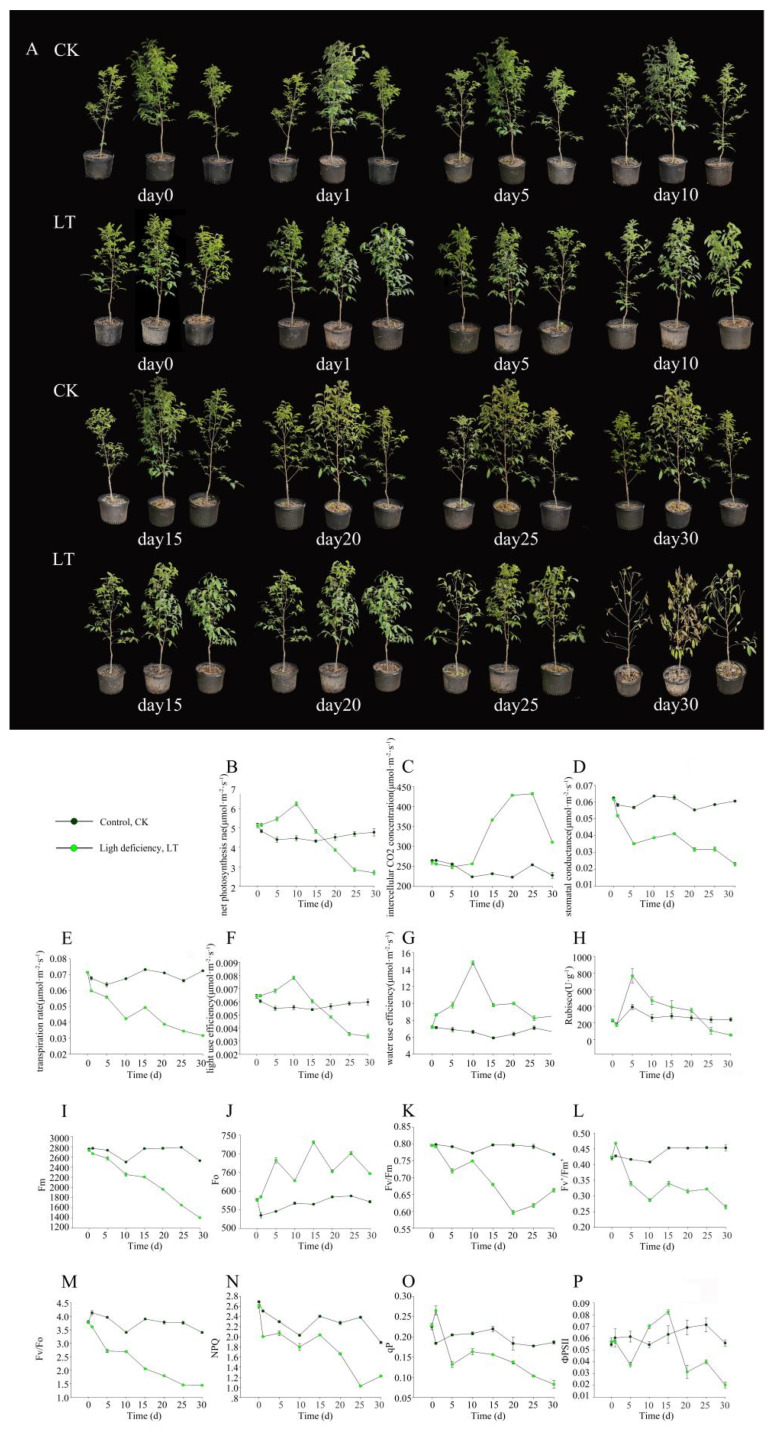
Phenotypic and physiological changes of *M. sinostellata* seedlings under light deficiency. (**A**) Phenotypic shifting of *M. sinostellata* during experiment. (**B**) Net photosynthesis rate, *Pn*. (**C**) Intercellular CO_2_ concentration, *Ci*. (**D**) Stomatal conductance, *Gs*. (**E**) Transpiration rate, *Tr*. (**F**) Light use efficiency, *LUE*. (**G**) Water use efficiency, *WUE*. (**H**) Rubiso activity. (**I**) Maximum Chl fluorescence yield obtained with dark-adapted leaf, Fm. (**J**) Minimum Chl fluorescence yield obtained with dark-adapted leaf, Fo. (**K**) Maximal photochemical efficiency, Fv/Fm. (**L**) Excitation energy capture efficiency of PSII, Fv’/Fm’. (**M**) Activity of PSII reaction centers, Fv/Fo. (**N**) Non-photochemical quenching, NPQ. (**O**) Photochemical quenching, qP. (**P**) Yield of PSII photochemistry, ΦPSII.

**Figure 2 plants-10-02261-f002:**
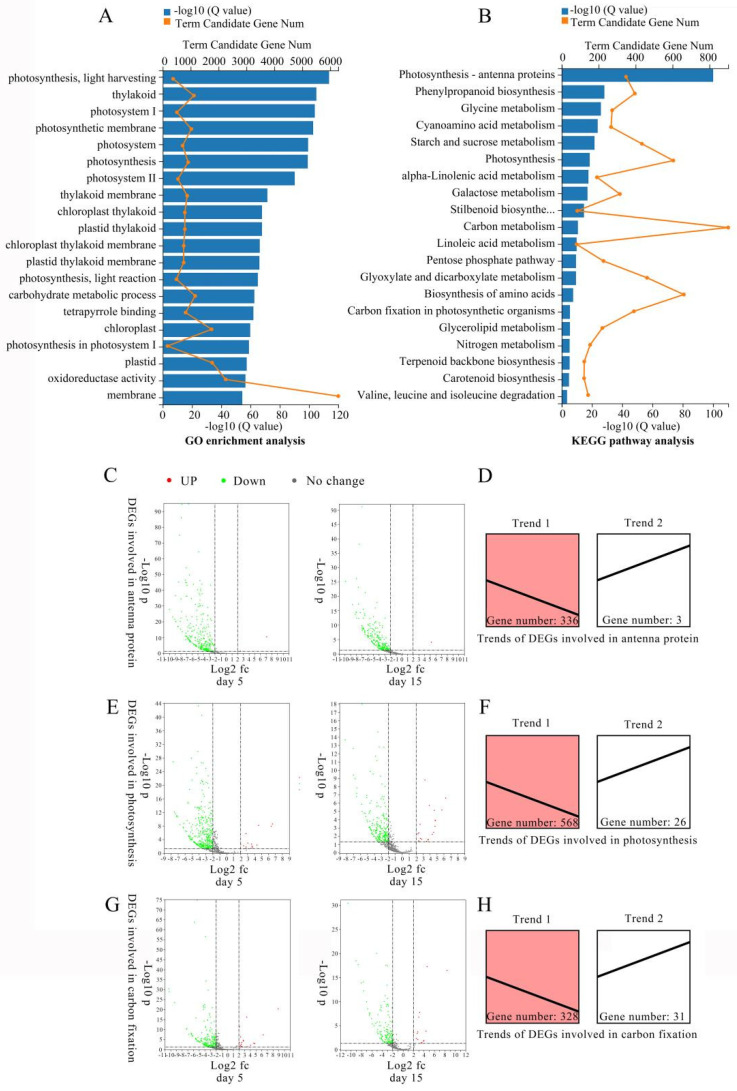
Analysis of DEGs involved in photosynthesis related pathways in *M. sinostellata*. (**A**) GO enrichment analysis of 22,433 DEGs. (**B**) KEGG enrichment analysis of 22,433 DEGs. The top 20 GO and KEGG Terms with the smallest Qvalue were selected to plot the chart, respectively. (**C**) Volcano plots of DEGs involved in antenna protein. (**D**) Trends of DEGs related to antenna protein. (**E**) Volcano plots of DEGs involved in photosynthesis. (**F**) Trends of DEGs related to photosynthesis. (**G**) Volcano plots of DEGs involved in carbon fixation. (**H**) Trends of DEGs related to carbon fixation.

**Figure 3 plants-10-02261-f003:**
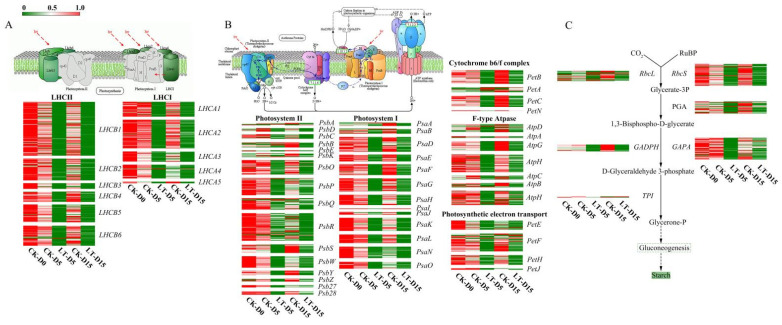
Heatmaps of DEGs involved in photosynthesis-related pathways in *M. sinostellata*. (**A**) Heatmaps of the expression patterns of genes involved in ‘photosynthesis-antenna protein’ pathway under light deficiency and normal light conditions. (**B**) Heatmaps of the expression patterns of DEGs participated in ‘photosynthesis’ pathway under light deficiency and normal light conditions. (**C**) Heatmaps of the expression patterns of DEGs related to ‘carbon fixation in photosynthetic organisms’ pathway under light deficiency and normal light conditions.

**Figure 4 plants-10-02261-f004:**
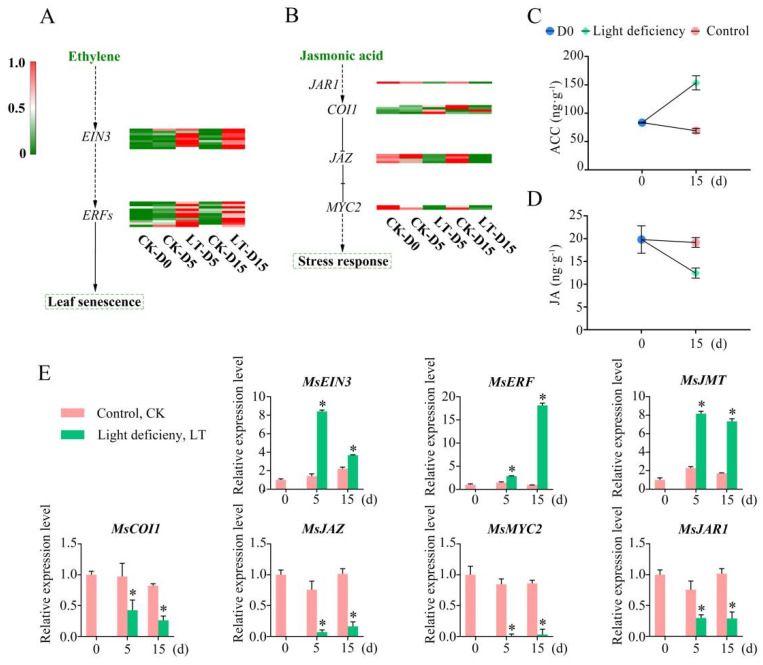
The impact of light deficiency on plant hormone concentration and signaling pathways. (**A**) Heatmap of genes involved in ethylene signal transduction under light deficiency and normal light conditions. (**B**) Heatmaps of genes involved in jasmonic acid signaling under light deficiency and normal light conditions. (**C**) Concentrations of ethylene at d0 and d15 under light deficiency and normal light conditions. (**D**) Concentrations of jasmonic acid at d0 and d15 under light deficiency and normal light conditions. (**E**) qRT-PCR analysis of key genes involved in plant hormone signaling under light deficiency and control conditions for 0 d, 5 d, and 15 d. Data are the means of three biological replicates and three technical replicates. The 2^−ΔΔct^ method was employed to conduct the gene differential expression analysis. * indicates significant differences in comparison with the control groups corresponding to time points at *p* < 0.05.

**Figure 5 plants-10-02261-f005:**
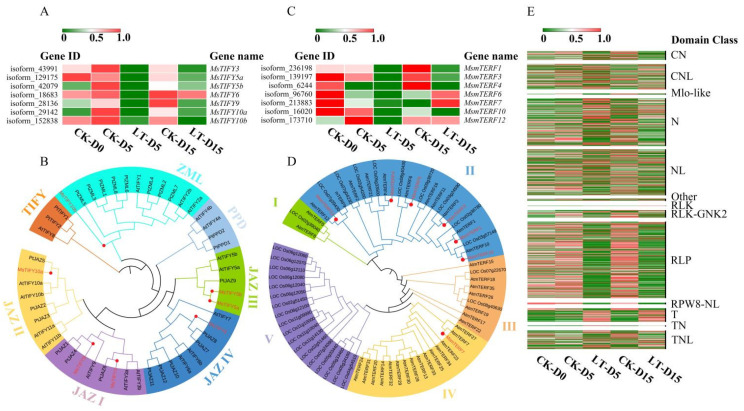
Analysis of TIFY family genes, mTERF family genes, and R-genes in *M. sinostellata*. (**A**) The expression profile of MsTIFY gene family in leaves of *M. sinostellata* under light deficiency and untreated conditions. Two pairs of duplicated paralogs are marked by lowercase letters. (**B**) Phylogenetic tree of TIFY protein sequences from *M. sinostellata*, Arabidopsis thaliana and Populus trichocarpa, which was constructed using the NJ (neighbor-joining) method with 1000 bootstrap replications. (**C**) Expression profiles of MsmTERFs under light deficiency and normal light conditions. (**D**) Phylogenetic tree of mTERF protein sequences from *M. sinostellata*, Arabidopsis thaliana, and Zea mays, which was constructed using the NJ method with 1000 bootstrap replications. (**E**) Expression patterns of 13 classification of R-genes in *M. sinostellata* under light deficiency and normal light conditions.

**Figure 6 plants-10-02261-f006:**
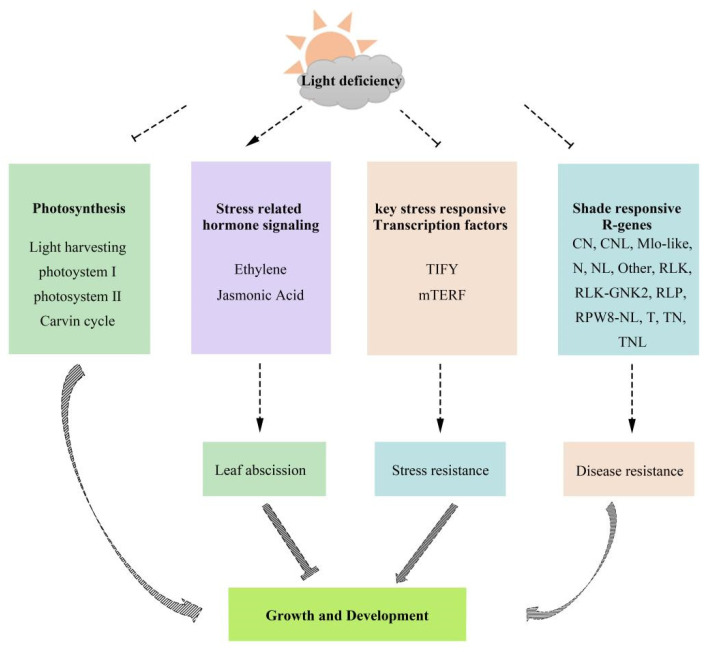
Hypothetical model of the light deficiency response mechanism in the endangered species *M. sinostellata*.

**Table 1 plants-10-02261-t001:** Classification of up- and down-regulated R-genes among the DEGs in *M. sinostellata*.

R-Gene Class	All R-Genes	Down-Regulated R-Genes	Up-Regulated R-Genes	Description
CN	40	28	12	Contains coiled-coil and NBS domains
CNL	109	50	59	Contains a central nucleotide-binding (NB) subdomain as part of a larger entity called the NB-ARC domain.
Mlo-like	5	4	1	Mlo-like resistant proteins
N	217	112	105	Contains NBS domain only, lack of LRR
NL	200	112	88	Contains NBS domain at N-terminal and LRR st the C-terminal, and lack of the CC domain
other	13	4	9	The class “Other” consists in a set of R proteins that do not fit into any of the known four classes, but that has resistance function.
RLK	1	1	0	RLKs, or Receptor like Kinases, consist of an extracellullar leucine-rich repeat region (eLRR) and an intracellular kinase domain.
RLK-GNK2	33	20	13	RLK class with additional domain GNK2
RLP	327	256	71	Receptor Like Proteins consists of a leucine-rich receptor-like repeat, a transmembrane region of ~25 AA, and a short cytoplasmic region, with no kinase domain
RPW8-NL	7	7	0	Contains NBS, LRR, and RPW8 domains
T	60	12	48	Contains TIR domain only, lack of LRR or NBS
TN	2	0	2	Contains TIR and NBS domains
TNL	72	38	34	Contains a central nucleotide-binding (NB) subdomain as part of a larger entity called the NB-ARC domain
Total	1086	644	442	

## Data Availability

The original transcriptome data used in this research was deposited in the National Center for Biotechnology Information (NCBI) with the accession number PRJNA770262.

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
