# Peer review of "Light Deficiency Inhibits Growth by Affecting Photosynthesis Efficiency as well as JA and Ethylene Signaling in Endangered Plant Magnolia sinostellata"

_plants, 2021, doi:10.3390/plants10112261_

Round 1

Reviewer 1 Report

The issues I pointed out have been sufficiently revised, and I think "Accept" is fine for this version.

Author Response

Response to Reviewer 1 Comments

Reviewer 1: The issues I pointed out have been sufficiently revised, and I think "Accept" is fine for this version.

Response: Thanks for the valuable comments. Your suggestions has helped us to make our manuscript more accurate and reasonable.

Reviewer 2 Report

The presented study explores the effects of light deficiency in the endangered plant Magnolia sinostellata. By analyzing photosynthetic performance and using comparative transcriptome analysis the authors provides insights in the light deficiency response mechanisms and proposed a model attempting to explain those observations. Main results include a reduction in light capture and photosynthesis and modification of phytohormones jasmonic acid and ethylene profiles. Also, the comparative transcription analysis allowed the identification of numerous core low-light responsive pathways.

In general, the manuscript is well written, displaying the results clearly and discussing them properly. The study was correctly designed and technically sound.

 Only some minor comments:

1- In the result section 2.1, between the fluorescence parameters it is informed the variable fluorescence “Fv”, which is the difference between maximal and basal fluorescence taken on dark-adapted leaves. Since this parameter is used to calculate other (Fv/Fm and Fv/F0), more informative ones, and since Fv is not very informative per se, I recommend the authors exclude it from the manuscript. Otherwise, the meaning of the parameter should be explicitly given.

2- In section 4.3, between the fluorescence parameters described appears ETR, but this is not informed in any other part of the manuscript so it should be suppressed from this section.

3- In lines 612-613 of the conclusion authors said “the electron transport chains in PS I and PS II were suppressed.”. First, the word suppressed sounds quite strong here and could be tone down, since there is a clear reduction but the electron transport is still functional. Moreover, from fluorescence data it is not possible to directly say nothing about PSI functionality, even when some suggestions could be made from them. To sum up, I would encourage the authors to modify this sentence.

Author Response

Response to Reviewer 2 Comments

Reviewer 2: The presented study explores the effects of light deficiency in the endangered plant Magnolia sinostellata. By analyzing photosynthetic performance and using comparative transcriptome analysis the authors provides insights in the light deficiency response mechanisms and proposed a model attempting to explain those observations. Main results include a reduction in light capture and photosynthesis and modification of phytohormones jasmonic acid and ethylene profiles. Also, the comparative transcription analysis allowed the identification of numerous core low-light responsive pathways.

In general, the manuscript is well written, displaying the results clearly and discussing them properly. The study was correctly designed and technically sound.

 Only some minor comments:

Point 1: In the result section 2.1, between the fluorescence parameters it is informed the variable fluorescence “Fv”, which is the difference between maximal and basal fluorescence taken on dark-adapted leaves. Since this parameter is used to calculate other (Fv/Fm and Fv/F0), more informative ones, and since Fv is not very informative per se, I recommend the authors exclude it from the manuscript. Otherwise, the meaning of the parameter should be explicitly given.

Response 1: Thanks for the comments. The Fv has been excluded from the manuscript and Figure 1.   

Point 2:  In section 4.3, between the fluorescence parameters described appears ETR, but this is not informed in any other part of the manuscript so it should be suppressed from this section.

Response 2: Thanks for the suggestions. The ETR related description has been deleted from this section, Line 641.

Point 3:  In lines 612-613 of the conclusion authors said “the electron transport chains in PS I and PS II were suppressed.”. First, the word suppressed sounds quite strong here and could be tone down, since there is a clear reduction but the electron transport is still functional. Moreover, from fluorescence data it is not possible to directly say nothing about PSI functionality, even when some suggestions could be made from them. To sum up, I would encourage the authors to modify this sentence.

Response 3: Thanks for your professional comments. The sentence in Line 631-632 has been modified: “Then, the electron transport chains in PS I and PS II can be affected.”

This manuscript is a resubmission of an earlier submission. The following is a list of the peer review reports and author responses from that submission.

Round 1

Reviewer 1 Report

1) The Introduction section needs some improvements. Important, well-known responses to shade in plants are not considered, such as the increase in the size of the light-harvesting antenna system or the role of phytochrome.

2) Materials & Methods:

- It should be indicated how PFD was measured (PAR?, equipment?, use of an integrating sphere?, …).

- Also, it is important to show how the light spectrum was modified by the net used to create shade conditions. The light in the understory of forest is not only a decrease in PFD; it is associated with a marked change in light quality with enrichment in far-red light. How do the authors take into account this aspect of natural shade in their study? Light supplied with far-red light could be used to simulate natural shade induced by surrounding plants.

3) Fig. 1. It is not clear at what PFD photosynthesis parameters, such as ΦPSII, qP and ETR, were measured. Since high light-grown plants and shade-exposed plants are acclimated to different PFDs, measuring photosynthesis at a single PFD is not very useful. Photosynthesis should be measured at the growth PFD or, preferentially, the light dependence of photosynthesis should be determined to characterize the effect of shade on the photosynthesis characteristics.

4) It is not surprising that transfer of Magnolia plants to shade induced changes in the expression of a large number of genes, including genes related to photosynthesis or hormones. Based on the transcriptomic data, the authors suggest that ABA-and ethylene-mediated signaling pathways are induced  whereas JA signaling is down-regulated. It might be useful to analyze hormones levels in the plants during shade exposure. It is also concluded from the transcriptomes that plant stress tolerance and resistance to diseases are down-regulated in the shade-exposed plants. These hypotheses should be tested by additional experiments, e.g. by testing plant resistance to pathogens and to stresses. 

5) A decrease in LHC gene expression in shade is in contradiction with typical shade responses observed in many species (increase in light harvesting). This difference should be discussed in more detail.

6) The Reference list should be revised: some journal abbreviations are wrong, pages are missing for several references, ….  

Author Response

Point 1: The Introduction section needs some improvements. Important, well-known responses to shade in plants are not considered, such as the increase in the size of the light-harvesting antenna system or the role of phytochrome.

Response 1: Thanks for the valuable comments and the valid suggestion. We have taken your advice and added the well-known shade responses including "the size of the light-harvesting antenna system and the role of phytochrome" in the introduction (Line 52-56, Line 72-77). The amendments could be found in the revision. In summary, we have added: "Phytochromes are the major light-sensing proteins and can sense changes in both light intensity and quality, among which phytochrome B plays a dominant role in shade avoidance responses including leaf expansion, stem elongation and early flowering(Line 52-56). "; "Light-harvesting antenna proteins have vital functions in the primary process in photosynthesis. To maximize photosynthetic efficiency, the size of the light harvesting antenna of the plants grown under low light intensities is often considerably larger than that grown under normal light[24]. However, the enlargement of the light-harvesting antenna is normally restricted within a certain size and as a result the energy generated by the photosynthesis reaction center is also limited under shading[25](Line 72-77). "

Point 2: Materials & Methods: It should be indicated how PFD was measured (PAR?, equipment?, use of an integrating sphere?, …).

Response 2: Thank you for the suggestions. We have revised the relevant description of the PAR determination method with more details of the light intensity measurement in the Materials & Methods section (Line 516-527) :"Throughout the experiment, these seedlings were placed in artificial climate room (photosynthesis active radiation (PAR) of 648 μmol·m-2·s-1, 14 h photoperiod, temperature 25°C, humidity 40-60%) in Zhejiang Agriculture & Forestry University. In order to simulate the shading condition of forest, seedlings in shade treatment group (shade treatment, ST) were placed in the shade set-up which was built using black shade net (25% of full light, PAR of 162 μmol·m-2·s-1, R/FR ratio: 1.09) and several bamboo poles (Figure S7). Seedlings in control group (control, CK) were not shaded (100% of full light, PAR of 648 μmol·m-2·s-1, R/FR ratio: 1.10). The illumination intensities in control group and shade treated group were measured in luminous flux (LUX) with a digital luxmeter (ZDS-10, Shanghai Jiading Xuelian Instrument Co., Ltd., Shanghai, China). Light intensity was converted from LUX to PAR following methods by Chen."

Point 3: Also, it is important to show how the light spectrum was modified by the net used to create shade conditions. The light in the understory of forest is not only a decrease in PFD; it is associated with a marked change in light quality with enrichment. in far-red light. How do the authors take into account this aspect of natural shade in their study? Light supplied with far-red light could be used to simulate natural shade induced by surrounding plants.

Response 3: Thanks for your professional comments. We have taken your advice and made revisions to address your concern in the revised manuscript. Indeed, the canopy shade changes both the light intensity and light quality to a low ratio of red light to far-red light in the understory. The light environment under canopy shade can be quite complex, containing low light intensity and varying light quality. To reduce confounding factors, like many previous studies performed in soybean, rice and Halimium halimifolium, we use black shade net to simulate natural shade conditions in this study, which mainly focused on the effect of low light intensity on plants. In addition, we have added data about light intensity and quality in different conditions in Table S8 (Control group, shade treated group, natural condition, canopy shade). The preliminary study suggested that M. sinostellata was hypersensitive to low light intensity and could severely impact on photosynthesis, stress tolerance and disease resistance of M. sinostellata plants. The light environment under canopy shade is quite complex. To control variables, we mainly investigated the impact of low light intensity on M. sinostellata. The effects of varying light quality on plants are also very significant, but it is clearly beyond the scope of the current study and warrants further research.

Point 4: Fig. 1. It is not clear at what PFD photosynthesis parameters, such as ΦPSII, qP and ETR, were measured. Since high light-grown plants and shade-exposed plants are acclimated to different PFDs, measuring photosynthesis at a single PFD is not very useful. Photosynthesis should be measured at the growth PFD or, preferentially, the light dependence of photosynthesis should be determined to characterize the effect of shade on the photosynthesis characteristics.

Response 4: In Figure 1, photosynthesis parameters of M.sinostellata in control group and shade group were measured using a LI-6400 photosynthesis analyzer with photosynthetic photon flow density (PPFD) set at 800 µmol·m-2·s-1, which has been described in Line 543. In the previous research of our research group, we found that the light saturation point of M. sinostellata is approximately 800 µmol·m-2·s-1[1]. Accordingly, it is reasonable to measure photosynthesis parameters with PPFD set at 800 µmol·m-2·s-1, which can scientifically reflect the changes in photosynthetic capacity of plants with or without shade treatment throughout the experiment.

Point 5: It is not surprising that transfer of Magnolia plants to shade induced changes in the expression of a large number of genes, including genes related to photosynthesis or hormones. Based on the transcriptomic data, the authors suggest that ABA-and ethylene-mediated signaling pathways are induced  whereas JA signaling is down-regulated. It might be useful to analyze hormones levels in the plants during shade exposure. It is also concluded from the transcriptomes that plant stress tolerance and resistance to diseases are down-regulated in the shade-exposed plants. These hypotheses should be tested by additional experiments, e.g. by testing plant resistance to pathogens and to stresses. 

Response 5: Thanks for the valulable comments and suggestions. The current study aims to identify the key factors affect plant growth as the result of shading through molecular approach focusing on transcriptome analysis. This research have provided novel insights into the shade response mechanism in an endangered ornamental tree species M. sinostellata through the analyses of transcriptome deep sequencing and photosynthesis efficiency. Shading directly reduced the intensity of light that was captured by light harvesting complex. Then, the electron transport chains and Calvin cycle were suppressed. The transduction patterns of of ABA, ET and JA were altered. The expression of stress responsive transcription factors and R-genes were suppressed. The revelation of ABA- and ethylene mediated signaling pathways playing a vital role in the shading management has directed our research towards the biochemical analysis of relevant plant hormones. Clearly it is beyond the scope of the current study to conduct biochemical analysis of plant hormones, but we agree with reviewer’s comments that it is necessary to conduct such studies in our subsequent research, to following up the findings as suggested by the transcriptome analysis in this manuscript.

Point 6: A decrease in LHC gene expression in shade is in contradiction with typical shade responses observed in many species (increase in light harvesting). This difference should be discussed in more detail.

Response 6: Thanks for your comments on this interesting finding. When the shade in-tolerant plants are exposure to short term shading stress, the size and gene expression of its light harvesting complex will increase in order to maximize light absorption for photosynthesis. However, the adaption range of light harvesting complex size is limited and so are the the energy generated by the photosynthesis reaction center under shading. However, when these plants are subjected to long term shading stress, the LHC gene expression will decreased, which are in accordance with the trend in M. sinostellata. The seemingly contradiction or discrepancy has been revised in the discussion (Line 425-430). in summary, "Under a short period of low light stress, the expression level of LHC genes in C. sinensis and soybean were significantly increased[15, 83]. In contrast, the expression of LHC genes in Arachis hypogaea showed a downward trend under long-term shading stress[11]. In M. sinostellata, the expression of LHCA1-5 and LHCB1-6 were also observed to decrease under long term shading."

Point 7: A The Reference list should be revised: some journal abbreviations are wrong, pages are missing for several references, ….

Response 7: We have carefully checked the reference list and corrected the errors. (Line 675-893).

Other edits:

Edit 1: Shading could change the light intensity and quality under nature, which are equally important. The light environment under canopy shade is quite complex. To control variables, we mainly investigated the impact of low light intensity on M. sinostellata and the research on light quality will be conducted in the near future. To make this view clearer, we have added following descriptions in the introduction and conclusion:

  1. "Canopy shade of evergreen plants in the upper layer of forest community will affect the growth of deciduous plants in the understory[8, 9], as it alters the light intensity as well as the light quality towards a lower ratio of red/far-red light (R/FR) in the forest community[10, 11], Line 48-51."
  2. "Changes in light quality with a low ratio of R/FR induce shade escape responses such as rapid stem elongation and increased apical dominance, which enable plants to reach above the upper vegetation to absorb sun light[14]. Low light intensity also alters chloroplast ultrastructure and photosynthetic metabolism[15]. Due to inadequate energy supply, long term exposure to low light conditions can limit growth in plants and even cause cellular damage in shade sensitive plants[16, 17], Line 56-62."
  3. "Previous studies have confirmed that shading is the main factor that affect the distribution and growth of deciduous magnolia in the understory of a forest community[4, 6]. In subtropical areas, the coniferous communities are favored by sinostellatabecause the light intensity in their understory is generally higher than that of evergreen broad-leaved communities due to the large gaps between needles[6]. The light environment under canopy shade can be quite complex, containing low light intensity and inconsistent light quality[63, 64]. To reduce the confounding factors, many studies focused on the impacts of low light intensity on a number of plant species, such as soybean[65], rice[66], and Halimium halimifolium[67], Line 130-141."
  4. "The preliminary study suggested that sinostellatawas hypersensitive to low light intensity and shading could severely impact on photosynthesis, stress tolerance and disease resistance of M. sinostellata plants. The light environment under canopy shade is quite complex. To control variables, we mainly investigated the impact of low light intensity on M. sinostellata, the effects of varying light quality under shade will be studied in further research, Line 506-512."

Edit 2: We have measured light intensity and quality under different conditions, Figure S8. The measurement method has been added in Line 527-529:"R/FR ratios under different conditions were measured by using a NIR spectrometer (Avaspec-HS-TEC, Avantes, The Netherlands), Table S8. "

Edit 3: In order to make the sentences of the article more concise, some sentences in the discussion has been simplified:

  1. Line 406-409, "Through a comprehensive analysis of transcriptome data, we found that shading had negatively impacted photosynthesis, stress tolerance and disease resistance of sinostellata."
  2. Line 461-464, "Through promoting the inactivation of PP2C and decrease in the suppression of serine/threonine-protein kinase SRK2 (SnRK2), the ABA-responsive element binding factors (ABFs) were phosphorylated[99]."
  3. Line 473-477, "The expression of JAR1decreased under shade, which interact with coronatine-insensitive protein 1 (COI1) and then leading to the degradation of JAZ proteins. The down regulation of JAZ under shade treatment is an indication of weakening of stress resistance[69]. As MYC2 is a key transcription activator of JA-Ile/COI1 signaling[103], its downregulation under shade treatment is perhaps not surprising. "
  4. Line 482-486, "TIFY and mTERFare two transcription factor families that are closely associated with defense and stress response[56, 105]. The TIFY family members such as GsJAZ2 and ZmTIFY11a were found to improve stress tolerance in max[55] and Zea mays[69], respectively. It has been reported that most TIFYs identified so far were inducible by unfavorable environmental conditions in order to enhance the stress tolerance in plants[55, 106]."

Reference

  1. Yu, Q. Title of Thesis. Effects of light and water on physiological characteristics and related gene expression of Magnolia sinostellataseedlings, Zhejiang Agriculture & Forestry University, Hangzhou, China,

Reviewer 2 Report

In this study, Lu et al. investigated the physiological and molecular mechanisms of shading stress in M. sinostellata. The results are well organized and I can't find any important issues for the publication.

Some minor points.

-L191 "A total of 22433 DEGs for shade response were identified". Which samples were compared to obtain this number of DEGs?

-Figure 5 Explain this figure in your manuscript. I can't find a citation for Figure 5 in the results or discussion.

-When an abbreviation appears at the beginning, the full name of the abbreviation should be included. For example, CK and ST in results.

-All figures are too small to properly understand what is being shown. Please make each figure larger so that it is easier to understand.

Author Response

Point 1: L191 "A total of 22433 DEGs for shade response were identified". Which samples were compared to obtain this number of DEGs?

Response 1: Thank you for the valid comments. We have made relevant revisions to explain the screening process of 22433 DEGs better, which could be found in the revision Line 225-227 :" In total, 11,850, 12,320, 7,165 and 15,389 DEGs were detected in CK-D0-vs-ST-D5, CK-D5-vs-ST-D5, CK-D0-vs-ST-D15 and CK-D15-vs-ST-D15, respectively (Figure S6A). After remove overlapping DEGs detected in the four comparison groups, a total of 22,433 DEGs for shade response were identified based on strict criteria...".

Point 2: Figure 5 Explain this figure in your manuscript. I can't find a citation for Figure 5 in the results or discussion.

Response 2: Figure 5 shows a shade response model of Magnolia sinostellata. (Now has been changed to Figure 6). We have carefully explained (Line 628-639) and cited (Line 639) Figure 6 in the conclusion:" Shading directly reduced the intensity of light that was captured by light harvesting complex. Then, the electron transport chains in PS I and PS II were suppressed. Due to the decline in ATP and NADPH production in photosystem, the carbon fixation in Calvin cycle and subsequent carbon metabolism were blocked. The enhanced signaling transduction of ABA, ET and MeJA accelerate leaf abscission, while the depressed JA signaling weakened stress tolerance in M. sinostellata. The decline in the expression of stress responsive transcription TIFY and mTERF also confirmed that stress resistance could have been impaired by severe shading. Most R-genes were downregulated by shading, indicating that plant resistance could also be declined. Overall, shading could severely impact on the growth and development of M. sinostellata plants through its direct effects on photosynthesis, stress tolerance and disease resistance."

Point 3: When an abbreviation appears at the beginning, the full name of the abbreviation should be included. For example, CK and ST in results.

Response 3: Thanks for your suggestions. Your advice has been taken and in the revision we have carefully checked throughout the entire manuscript and added the full name with the first abbreviation. Specific edits has made as follows:"NADPH" has been changed to "Nicotinamide Adenine Dinucleotide Phosphate" (Line 79); "ATP" has been changed to "Adenosine-Triphosphate" (Line 79); "ST" has been changed to "shade treatment" (Line 155); "CK" has been changed to "control" (Line 155); "Pn" has been changed to "net photosynthetic net (Pn)" (Line 167); "Ci" has been changed to "intercellular CO2 concentration (Ci)" (Line 167); "Gs" has been changed to "stomatal conductance (Gs)" (Line 168); "Tr" has been changed to "transpiration rate (Tr)" (Line 168); "Fm" has been changed to "Maximal fluorescence (Fm)" (Line 177); "Fv/Fm" has been changed to "maximal quantum yield of PSII (Fv/Fm)" (Line 177); "Fv’/Fm’" has been changed to "excitation energy capture efficiency of PSII (Fv’/Fm’)" (Line 178); "Fv" has been changed to "variable fluorescence (Fv)" (Line 178); "qP" has been changed to "photochemical quenching (qP)" (Line 179); "Fv/Fo" has been changed to "active PSII reaction centers (Fv/Fo)" (Line 179); "NPQ" has been changed to "non-photochemical quenching (NPQ)" (Line 180); "Fo" has been changed to "initial fluorescence (Fo)" (Line 182); "LHC I" has been changed to "Light harvesting complex I (LHC I, LHCA1-5)" (Line 265); "LHC II" has been changed to "Light harvesting complex II (LHC II, LHCB1-7)" (Line 266).

Point 4: All figures are too small to properly understand what is being shown. Please make each figure larger so that it is easier to understand.

Response 4: We have adjusted the size of each figure and the size of the text in the figure to make it clear to understand what is being shown: Figure 1 (Line 185); Figure 2 (Line 242); Figure 3 (Line 250); Figure 4 (Line 300); Figure 5 (Line 343); Figure 6 (Line 641).

Other edits:

Edit 1: Shading could change the light intensity and quality under nature, which are equally important. The light environment under canopy shade is quite complex. To control variables, we mainly investigated the impact of low light intensity on M. sinostellata and the research on light quality will be conducted in the near future. To make this view clearer, we have added following descriptions in the introduction and conclusion:

  1. "Canopy shade of evergreen plants in the upper layer of forest community will affect the growth of deciduous plants in the understory[8, 9], as it alters the light intensity as well as the light quality towards a lower ratio of red/far-red light (R/FR) in the forest community[10, 11], Line 48-51."
  2. "Changes in light quality with a low ratio of R/FR induce shade escape responses such as rapid stem elongation and increased apical dominance, which enable plants to reach above the upper vegetation to absorb sun light[14]. Low light intensity also alters chloroplast ultrastructure and photosynthetic metabolism[15]. Due to inadequate energy supply, long term exposure to low light conditions can limit growth in plants and even cause cellular damage in shade sensitive plants[16, 17], Line 56-62."
  3. "Previous studies have confirmed that shading is the main factor that affect the distribution and growth of deciduous magnolia in the understory of a forest community[4, 6]. In subtropical areas, the coniferous communities are favored by sinostellatabecause the light intensity in their understory is generally higher than that of evergreen broad-leaved communities due to the large gaps between needles[6]. The light environment under canopy shade can be quite complex, containing low light intensity and inconsistent light quality[63, 64]. To reduce the confounding factors, many studies focused on the impacts of low light intensity on a number of plant species, such as soybean[65], rice[66], and Halimium halimifolium[67], Line 130-141."
  4. "The preliminary study suggested that sinostellatawas hypersensitive to low light intensity and shading could severely impact on photosynthesis, stress tolerance and disease resistance of M. sinostellata plants. The light environment under canopy shade is quite complex. To control variables, we mainly investigated the impact of low light intensity on M. sinostellata, the effects of varying light quality under shade will be studied in further research, Line 506-512."

Edit 2: We have measured light intensity and quality under different conditions, Figure S8. The measurement method has been added in Line 527-529:"R/FR ratios under different conditions were measured by using a NIR spectrometer (Avaspec-HS-TEC, Avantes, The Netherlands), Table S8. "

Edit 3: In order to make the sentences of the article more concise, some sentences in the discussion has been simplified:

  1. Line 406-409, "Through a comprehensive analysis of transcriptome data, we found that shading had negatively impacted photosynthesis, stress tolerance and disease resistance of sinostellata."
  2. Line 461-464, "Through promoting the inactivation of PP2C and decrease in the suppression of serine/threonine-protein kinase SRK2 (SnRK2), the ABA-responsive element binding factors (ABFs) were phosphorylated[99]."
  3. Line 473-477, "The expression of JAR1decreased under shade, which interact with coronatine-insensitive protein 1 (COI1) and then leading to the degradation of JAZ proteins. The down regulation of JAZ under shade treatment is an indication of weakening of stress resistance[69]. As MYC2 is a key transcription activator of JA-Ile/COI1 signaling[103], its downregulation under shade treatment is perhaps not surprising. "
  4. Line 482-486, "TIFY and mTERFare two transcription factor families that are closely associated with defense and stress response[56, 105]. The TIFY family members such as GsJAZ2 and ZmTIFY11a were found to improve stress tolerance in max[55] and Zea mays[69], respectively. It has been reported that most TIFYs identified so far were inducible by unfavorable environmental conditions in order to enhance the stress tolerance in plants[55, 106]."
